# Tell Me Where to Go: A Composable Framework for Context-Aware Embodied Robot Navigation

**Harel Biggie**
Department of Computer Science
University of Colorado Boulder United States
`harel.biggie@colorado.edu`

**Ajay Narasimha Mopidevi**
Department of Computer Science
University of Colorado Boulder
`ajay.mopidevi@colorado.edu`

**Destin Woods**
Department of Computer Science
University of Colorado Boulder
`destin.woods@colorado.edu`

**Christoffer Heckman**
Department of Computer Science
University of Colorado Boulder
`christoffer.heckman@colorado.edu`

**Abstract:** Humans have the remarkable ability to navigate through unfamiliar environments by solely relying on our prior knowledge and descriptions of the environment. For robots to perform the same type of navigation, they need to be able to associate natural language descriptions with their associated physical environment with a limited amount of prior knowledge. Recently, Large Language Models (LLMs) have been able to reason over billions of parameters and utilize them in multi-modal chat-based natural language responses. However, LLMs lack real-world awareness and their outputs are not always predictable. In this work, we develop a low-bandwidth framework (NavCon) that solves this lack of real-world generalization by creating an intermediate layer between an LLM and a robot navigation framework in the form of Python code. Our intermediate layer shoehorns the vast prior knowledge inherent in an LLM model into a series of input and output API instructions that a mobile robot can understand. We evaluate our method across four different environments and command classes on a mobile robot and highlight our framework's ability to interpret contextual commands.

**Keywords:** Natural language, navigation, contextual navigation

## 1 Introduction

Humans have the remarkable ability to navigate through unfamiliar environments, e.g., in a town or through a building, by relying solely on our priors and descriptions of the environment [1]. Motivated by the difficulties in directing robots in collaborative teams with humans such as those used in search and rescue [2, 3, 4, 5] operations, we aim to develop a framework that allows humans to provide high-bandwidth instructions to robots in the form of natural language.

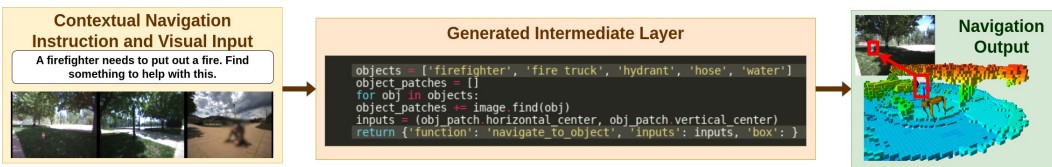

Figure 1: Contextual navigation example for a firefighting situation. We see our compositional framework generates code for the robot to both find and then plan to a fire extinguisher.

To achieve this, it is generally required that a robot associate natural language utterances to the physical world using sensing modalities onboard the robot, a process known as grounding. However,

7th Conference on Robot Learning (CoRL 2023), Atlanta, USA.

unlike humans, robots are not currently capable of integrating prior experiences into a vast wealth of priors to aid this association.

Recently, Large Language Models (LLMs) [6] have been able to reason over billions of parameters and utilize the results in domains such as dialogue-based responses [7, 8, 9], code generation [10, 11, 12], and multimodal visual question and answering (VQA) tasks [13]. This type of unsupervised learning shows incredible potential for the generalization of prior knowledge but these models lack real-world experience. A lack of physical experience is perhaps one reason why transitioning techniques developed through research in VQA and view-based navigation into embodied agents has not kept pace with the research in these fields.

Additionally, with the current generation of LLMs, it's inherently challenging to interpret the rationale behind the outputs that are generated by the model. In this work, we address both the grounding and transparency issues using a composable framework designed for robot navigation. Our pipeline extends the reasoning framework for visual inference presented in [14] to an embodied robotic agent. With our composable framework, we leverage LLM priors (GPT-3.5), state-of-the-art object detectors [15], and classical robotic planning algorithms [16, 17, 18] to perform zero-shot natural language based navigation in four unique environments.

Specifically, our contributions are as follows:

- We extend the concept of modular neural networks and define a new framework for composable robot navigation which we call NavCon. Our framework requires a minimal uplink for the robot since all of our planning, mapping, and localization is performed onboard.
- We evaluate different 2D input representations to determine an effective way to extract spatial and conceptual knowledge from LLMs.
- We perform extensive real-world experiments in a variety of environments and show that our framework is able to navigate to landmarks based on natural language. Furthermore, the framework is able to deduce appropriate navigational goals from the context of a sentence.

## 2  Related Works

**Grounding Language:** Associating natural language with an embodied agent requires grounding utterances to the physical world in which the robot operates. Various approaches have been used to associate language with the physical domain ranging from probabilistic graph-based structures [19, 20, 21] to end-to-end learning-based methods [22]. Graph-based approaches have shown promising results but their generalizability is limited to a fixed training corpus. On the other hand, LLMs have proven to be adept at reasoning over large amounts of unsupervised training data [23, 24] using transformer-based backends [25]. These models also known as foundation models due to their comprehensive knowledge have recently been leveraged for task and motion planning using reinforcement learning on a set of skills [26], and in an end-to-end fashion [22] with object scene representation transformers [27]. While these approaches have made remarkable strides they still suffer from the explainability problem and further they don't share any of the path planning guarantees that traditional planning methods share [28].

**Modular Neural Network Frameworks and Code Synthesis:**  Modular Neural Networks [29, 30, 31] are remarkably adept at answering questions about images in a task commonly known as VQA [32]. Until recently many of these methods were limited in their ability to generalize to other domains due to the difficulty of generating interfaces between modules. In [14] this was addressed by using recent code generation techniques [33] from LLMs to create these interface layers. Specifically, these modular frameworks enable the generation of code that dictates the interaction between robust object detectors [15, 34, 35, 36, 37], depth estimators [38, 39, 40]. What's more remarkable is that these LLMs can also utilize existing functions inside of the Python language such as sorting and conditionals without any additional training [41, 42, 12, 24]. Based on these exceptional findings we aim to extend the use of these modular concept-learning style frameworks to the robotic navigation domain.

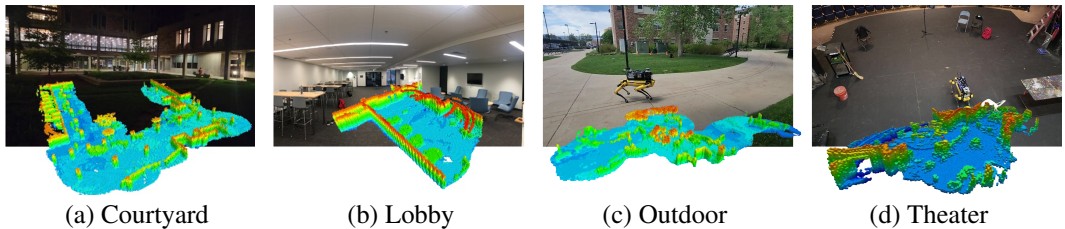

| (a) Courtyard | (b) Lobby | (c) Outdoor | (d) Theater |

Figure 2: Full volumetric maps and representative images from the four testing environments.

**Path Planning for Robotic Navigation:** Path planning for robotic navigation and exploration is a long-studied research area [43, 44] with solutions ranging from sampling-based methods[16, 45] to graph-based lattice structures [46]. Variations of these probabilistic approaches are being used for self-driving cars [47], exploration [18, 4, 3, 5] in complex environments. These approaches benefit from probabilistic guarantees on convergence and optimality [16] and adaptability to new environments making them natural choices to utilize in language grounding framework.

**Embodied Language-Based Navigation:** Recently a few other works have leveraged foundation models for embodied navigation. Clip on Wheels (CoW) is proposed in [48] which combines text and image caption models [49] with frontier-based exploration algorithms [50] to perform object exploration in simulation. In [51] landmark-based navigation is performed using pre-trained visual [48], language, and navigation modules [52]. Visual Language Navigation Maps (VLMaps) [53] learn a spatial language representation by combining RGB video feeds, code generation, and foundation models. SayCan [54], and Palm-E [22] utilize the PALM [55] language model to perform skill-based navigation and end-to-end navigation respectively. In general, these methods are not tested on contextual examples, and some require the preprocessing of maps; in contrast, we test on situated examples with an experimental platform and require no situated priors.

## 3 Methodology

We develop a framework for Navigation with Context (NavCon) that leverages the rich contextual priors of LLMs and creates an executable code layer that interfaces with planning algorithms running on an embodied agent. Drawing on paradigms from modular neural networks [56, 57, 14], and concept learning techniques [58], we define a modular system that takes inputs at various stages to fully define a final navigational output for our system. This enables the use of arbitrary intermediate and substitutable layers which can also be executed on a distributed computing infrastructure. Formally, we have a collection of inputs $\psi$ composed of a visual input $v \in V$ an RGB image or multi-perspective collection of images, $m \in M$ a volumetric map of the space to be navigated, and $c \in C$ a natural language navigation command. Our system takes initially as input a command $c$ to generate code $\gamma_c$. This code consumes $v$ in order to resolve the grounding problem, i.e., to determine which object is being referred to in $c$ for navigational instructions, as well as $m$ in order to emplace the object into the world around the robot. This outputs $i = \theta(v, m|\gamma_c) \in \mathbb{R}^3$, a 3D waypoint to which navigation will commence in order to output $\mathcal{P}$ a robot trajectory which is a continuous function in $\mathbb{R}^3$. A graphical overview of the framework can be found in Figure 3.

**Input Representations:** Our visual input $v$ is in the form of an RGB image, either a semi-panoramic view or three separately labeled spatial images (left, front, right). Typical inference methods over RGB images [36, 35, 15] reason in 2D over the image. For embodied navigation, we need to reason in 3D and associate different camera viewpoints with their associated 3D spatial relations. Embodied navigation introduces spatial relations that are difficult to reason over in 2D such as "behind", "in front of", or "on your right". To determine an appropriate input representation we evaluate the difference between sending in a concatenated image of all viewpoints (semi-panoramic) from the agent or sending in each frame separately with a spatial definition, i.e., right, front, or left. We

find that spatial reasoning is best performed on a concatenated image and we present the details in Section 4.

**Intermediate Layer:** We generate $\theta$ using recent advances in code generation models [10, 42]. We provide a functional set of navigation instructions in the form of a Python API for the code generation model. The full prompt containing the API can be found in the Appendix. These instructions include specifications for how to perform visual inference on the input $v$ using a similar paradigm found in [14]. These API instructions include directions for finding an object, checking if an object has a certain property, etc. Additionally, we provide an API specification for interfacing with our geometric planner, resulting in code generated based on the natural language prompt $\gamma_c$.

**Planning:** We use a graph-based planner first presented in [18, 4] where each sample is a robot position parameterized by the robot's width and length. During sampling, the collision checks are performed by projecting the samples along the path to the ground on the map $m$ below the sample. The elevation change at the footprint of the sample is calculated and if these differences exceed a given threshold or there are not enough sampled points (indicating a hole in the ground) the path is marked invalid and not added to the graph. We create a navigational API that takes in the center coordinate $p$ of an object in image space and projects it as a landmark $l$ on a 3D map. Using this strategy, a graph of plans $\mathcal{G}$ is constructed by sampling points parameterized by the robot's width and length [16]. Additional semantic labels are added to the map for staircases as described in [59] using normal estimation on point clouds. When a 3D waypoint is input to the planner we select the best path $\mathcal{P}$ from $\mathcal{G}$, for the robot to follow. If no path exists, we plan to the boundary of the graph and resample until the goal is reached.

Waypoints are passed to the planner using the output $i$ from the intermediate layer. Specifically, image coordinates are translated into waypoints by executing $\theta$ and associating the result onto a 3D map $m$ [60] using ray casting. The map $m$ is generated online using [61] and the translation layer $\theta$ creates the necessary code to translate between the inputs $c$, $v$, $m$ as and the output $i$. We then plan to the waypoint by selecting the best path from $\mathcal{P}$.

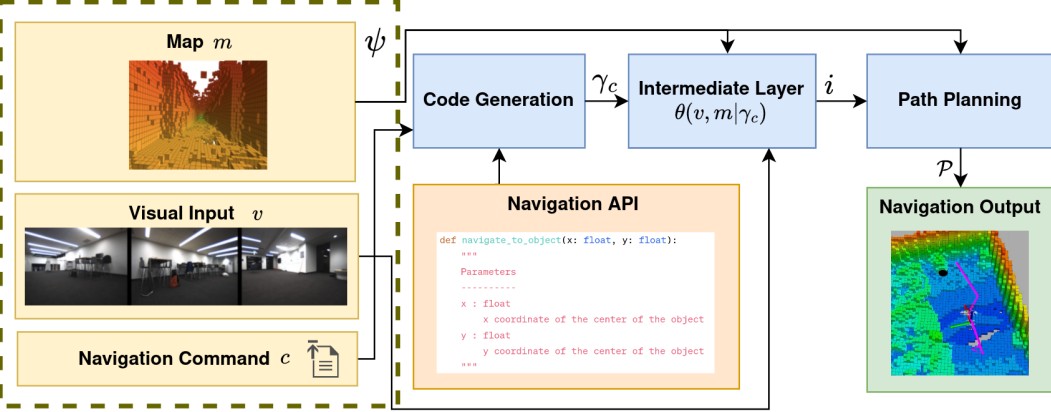

Figure 3: A graphical overview of the full system framework.

# 4 Experimental Results

We run two sets of experiments on a Boston Dynamics Spot equipped with a custom sensing suite consisting of a 3D 64 beam Ouster lidar, an IMU, and 3 RGB cameras providing a semi-panoramic view of the environment. Our first experiment is designed to determine the best input representation for the visual layer and the second set tests the ability of our system to perform navigation in a wide variety of real-world environments.

For all of our experiments, we leverage ideas from human concept learning as in [58] to categorize our sentences into four categories: Generic, Specific, Relational, and Contextual. Generic sentences

are simply sentences that imply "Go to something", e.g. "Walk to the backpack.". Specific sentences include a distinguishing piece of language such as a color attribute that directs the robot to one specific object in the scene. For example, if there are two backpacks, one red, and one black, then a specific sentence would be "Drive to the black backpack." Relational sentences are any sentences that describe spatial relationships between objects in the scene such as, to the right of, on top of, etc. An example in this category is "Move to the backpack on top of the chair." Contextual examples, require the robot to interpret the navigational goal based on background information. For instance, the sentence "Find something that can carry water." requires the robot to know that a cup or a bucket can hold water.

We generated sentences by eliciting verbs for navigational policies from an LLM (GPT 3.5-0301). Then objects in the scene were inventoried, and sentences were constructed based on verb-object combinations as well as their properties and relationships to other objects in the environment. These sentences were then diversified into contextual commands by providing reference to the objects' attributes or affordance and our full list of sentences can be found in the Appendix.

We evaluate the success of the method based on the following criteria. The code generation step is considered successful if the generated intermediate layer $\gamma_C$ is both syntactically and logically correct. In other words, if every component worked successfully then the generated code would solve the navigation command. Object Detection (OD) is considered successful if the correct object is extracted from the visual input in the intermediate layer as visualized by the bounding box. Similarly for waypoint projection (WP) if the 3D projection is in the correct location on the map $m$ this step is considered a success. The planning and execution step is considered successful if the robot plans and then traverses to within 0.2m of the target object. For all tables percentages are calculated based on the total number of evaluated sentences.

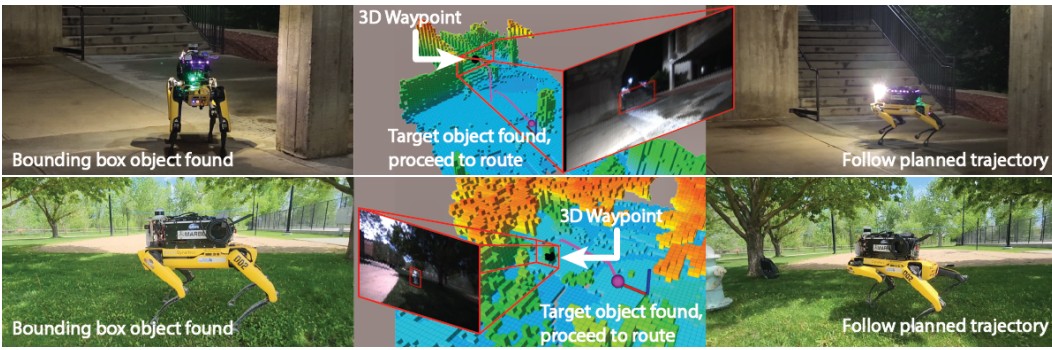

Figure 4: Example sentences, visual detections, maps, and planned paths.

Encoding spatial relationships into a model without a concept of the physical world presents significant challenges. We determine an effective scheme for inputting three different viewpoints into the framework through empirical evaluations. Take the sentence, "go to the chair on the right" implies only the right half of the total FOV should be investigated but "Go to the backpack that is to the right of the chair." could be in any image. To solve these limitations we use two different sets of input representations, the first is all three images stitched together with padding in between frames (**A**). In this case, we instruct the LLM on the order of the images (left, front, right) and let the model handle the spatial reasoning. In the second case, we process the frames individually (**B**) and let the model decide which frames to look at. We explicitly prompt the model with example code snippets that solve the difference between "to your right" and "right of" styles of language. at configuration. In this set of experiments, we evaluate the correctness of the generated code as well as if the code produced the correct object in the map $m$. This scene was created in a classroom setting with good lighting conditions as the emphasis was to evaluate the effect of the input representation. Results for successful code generation and object detections for the two input schemas are summarized in Table 1.

We also compare our results to OFA [62], a sequence-to-sequence multi-modal model finetuned on RefCOCO [63] expressions for the object detection stage. An additional baseline is performed where we utilize an LLM (GPT-3.5-0301) to convert a navigational command to a referring expression such as the ones in RefCOCO. Our full prompt for this pipeline is provided in the Appendix.

| Category | Count | A (%) | B (%) | OFA (%) | OFA+LLM (%) |
|---|---|---|---|---|---|
| Generic | 12 | 100 | 100 | 25.0 | 50.0 |
| Specific | 12 | 91.67 | 66.67 | 25.0 | 50.0 |
| Relational | 15 | 86.67 | 53.33 | 6.67 | 13.33 |
| Contextual | 11 | 81.82 | 45.45 | 0 | 36.37 |
| Total | 50 | **90** | 66 | 14 | 36 |

Table 1: Results showing the difference between the concatenated input representation **A** and the sequential representation **B**.

From Table 1 we find that concatenating images significantly outperforms sending in the three frames separately. As expected, in the generic navigation case we are able to achieve 100% success in generating the intermediate layer and for object identification. We see that the individual frame configuration (**B**) begins to break down when reasoning over specific objects. Specifically, the model fails to reason over spatial relations and object ordering when objects appear in more than one frame. For example, taking the sentence "Go to the middle outlet" only works when the three outlets are present in the same camera frame. If the middle outlet is in the front frame but the right outlet is in the right camera frame, this method fails. We explicitly see this in the generated code:

```
outlet_patches.sort(key=lambda x: x.horizontal_center)
middle_outlet = outlet_patches[len(outlet_patches) // 2]
```

where each detected outlet is ordered based on its horizontal coordinate in the image. Since all outlets are merged into the same list the images have overlapping coordinates causing these failures. Configuration **B** struggles with relational sentences between objects for similar reasons.

In our second set of experiments, we evaluate the ability of our framework to navigate in real-world environments. The four environments shown in Figure 2 range from an indoor lobby setting to a dark outdoor courtyard. We select these environments to highlight our framework's ability to generalize in multiple scenarios. In each environment, we test multiple sentences across the four categories (see the Appendix for the full list of sentences). Sentences are primarily based on landmarks already present in the environment but in some cases, we add additional artifacts to enable more extensive testing.

| Category | Count | Code(%) | OD(%) | WP(%) | Path& Exec(%) | OFA (%) | OFA+ LLM (%) |
|---|---|---|---|---|---|---|---|
| Generic | 22 | 100 | 81.82 | 68.18 | 68.18 | 13.64 | 22.73 |
| Specific | 19 | 89.47 | 89.47 | 78.95 | 73.68 | 0 | 15.79 |
| Relational | 44 | 72.73 | 59.10 | 59.10 | 59.10 | 18.19 | 36.37 |
| Contextual | 29 | 65.52 | 41.38 | 41.38 | 41.38 | 6.90 | 24.14 |
| Total | 114 | 78.95 | 64.04 | 59.65 | **58.77** | 11.40 | 27.19 |

Table 2: Summary of category-wise success rate for each of the four command types.

We find that we are able to successfully generate code for navigational plans using a variety of verbs e.g., walk, go, drive, run, etc by leveraging the rich vocabulary knowledge present in foundation models. In fact, we can even say phrases like "sashay to the stop sign" which will be interpreted as a "go to" style navigation command.

From Tables 2 and 3, we see that for generic objects our framework has a 100% success rate for code generation. In this case, code is manually evaluated and deemed successful if the generated

| Scenes | Count | Code(%) | OD(%) | WP(%) | Path& Exec(%) | OFA (%) | OFA+ LLM (%) |
|--------|-------|---------|-------|-------|---------------|---------|--------------|
| Theater | 30 | 90 | 70 | 66.67 | 63.33 | 3.33 | 13.33 |
| Lobby | 29 | 65.52 | 48.28 | 44.83 | 44.83 | 13.79 | 24.14 |
| Outdoor | 24 | 87.5 | 79.17 | 70.83 | 70.83 | 12.5 | 33.33 |
| Courtyard | 31 | 74.19 | 61.29 | 58.06 | 58.06 | 16.13 | 38.71 |
| Total | 114 | 78.95 | 64.04 | 59.65 | **58.77** | 11.40 | 27.19 |

Table 3: Summary of scene-wise success rate for each of the four command types.

code would produce the correct results assuming all the rest of the composed modules were executed successfully. In both tables *Code* represents the success of the code generation step, *OD* represents the success of the object detector (GLIP), *WP* is the success of the 3D waypoint projection and *Path&Exec* is the success of the graph-based planning and navigation module. Our failures occurred in the object detection module where either the wrong object was detected or the object was not detected at all. However, since our framework is modular, the object detector can be substituted and as state-of-the-art vision detectors improve our framework will improve in turn. Additional failures occur in the waypoint projection step. These occurred for two reasons. In the first case, the vision system detected further out than the map resulting in projections at the edge of the map rather than the actual object. For the second case, projections missed the correct voxel on the map. This was either due to a ray "clipping" a closer object in the projection process. For longer projections (10–12m), the camera-to-lidar calibration caused enough error to make the projection inaccurate, which could be resolved through mean point clustering. We also note that we observed a 100% success rate in following all planned paths.

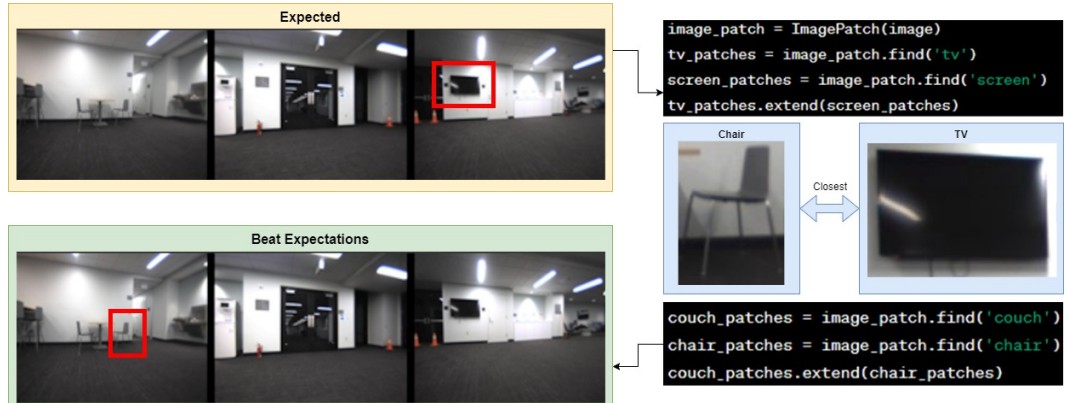

Figure 5: Results for the sentence "Find me somewhere to watch a movie."

**Selected Qualitative Examples:**

Leveraging the vast knowledge base encoded into LLMs enables a degree of spatial reasoning and the ability to infer navigation instructions based on the context of the sentence. For example, take the sentence "Find me something to help a firefighter" as shown in Figure 1 requires the robot to identify a list of objects that could help a firefighter and then look for them. From the snippet of the generated code, we see that the robot successfully looks for fire extinguishers and fire hydrants, both of which would aid a firefighter. Additionally, we are able to ask the robot to "Find something to clean up a mess" and it will find a mop or a broom.

What's more remarkable than simply interpreting correct navigation commands from context is our framework's ability to leverage foundation models to find more refined solutions to a task in some instances. Take for example the sentence "Find me somewhere to watch a movie," we would expect the robot to go and find a TV or some kind of screen. Of course, the first object the robot looks for

is the screen but more remarkably the robot also looks for a chair which is closest to the screen. We can see this in Figure 5 where the generated layer $\theta$ explicitly looks for "TVs, screens" and then looks for "chairs" and "couches."

As powerful as the contextual engine (GPT-3.5) is in our framework, it still has limitations when interpreting the physical world. In Figure 6 we tell the robot to "Run upstairs." and to no surprise, it will successfully generate code and plan up the stairs. However, when given the utterance "Go to the second floor." the following code is generated:

```
floor_patches = ImagePatch(image).find('floor')
floor_patches.sort(key=lambda x: x.vertical_center)
if len(floor_patches) < 2:
    return {'function': 'None', 'error': 'Image does not contain at least
        two floors.'}
second_floor_patch = floor_patches[1]
```

This code is nonsensical to the actual task because it's a literal interpretation of the sentence. Counting floors has nothing to do with going to the second story of a building. Examples like these highlight the need for additional methods of encoding 3D spatial reasoning into natural language-based embodied navigation frameworks.

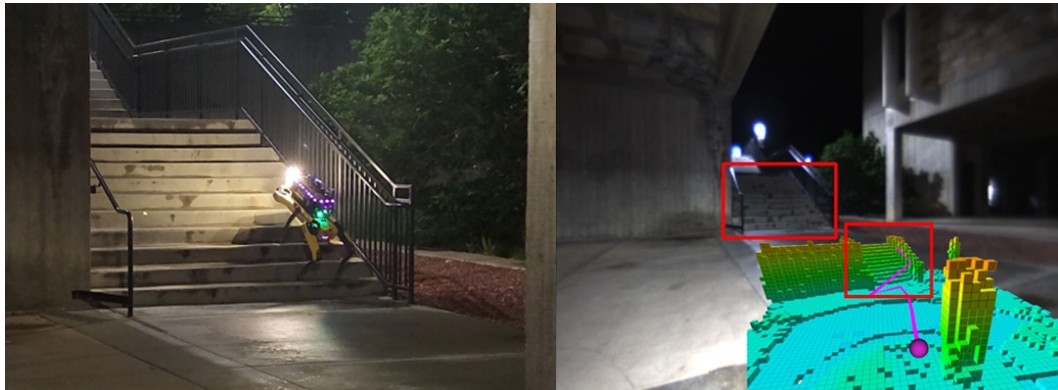

Figure 6: Scene used to tell the robot to "Run upstairs" and "Go to the second floor". The second sentence fails due to the code trying to "count floors"

## 5  Conclusion and Limitations

Our composable framework shows remarkable accuracy and performance in generating code that fits the navigation task. In controlled experiments, we achieve 90% accuracy across 50 different sentences. When taken to the real world the, model still performs exceptionally well with zero additional input. This is highlighted by a 70% execution accuracy in large outdoor environments.

While this work makes significant strides toward embodied navigation, we observe many opportunities for future effort. Namely, operating in only 2D space creates challenges for spatial relationships such as "behind" the robot or "to the right of." Additionally, we are limited by the visual range of the robot: therefore 3D projections need to be exponentially more accurate for both the smaller and further away an object is in the given input image. Large multimodal foundation models are promising, but promise uncertain runtime performance given their lack of availability and scalability. Finally, the current implementation of the framework is limited to landmark-based aviation and future work includes extending this framework to include more navigational inputs such as unbounded exploration in a direction or, having the robot explore until a condition is met. We also wish to include a module that operates on 3D maps and a module that supports sequential instructions.

**Acknowledgments**

This work was supported through the DARPA Subterranean Challenge cooperative agreement HR0011-18-2-0043, the National Science Foundation #1764092 and #1830686, and USDA-NIFA #2021-67021-33450.

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

# 6 Appendix

## 6.1 Environment Descriptions

We test our framework on four different real-world environments. Further details about each of the environments are listed below:

- **Theater:** an indoor environment simulating a stage-like environment during set construction.
- **Lobby:** an indoor environment, mostly occupied with chairs and tables, posing more challenges in distinguishing specific objects and planning
- **Outdoor:** a larger environment with objects such as trees, cars, bikes, and sporting areas.
- **Courtyard:** an outdoor environment tested at night to challenge the framework with low-light images.

Along with the general objects present in the scene, we placed some additional items like backpacks, shoes, and cones into the scene. All the prompts tested in these 4 different environments are presented in Tables 5, 6, 7, and 8. For each of the prompts, we also highlighted whether each step succeeded or not.

## 6.2 OFA Comparisions

The full prompt for the OFA [62]+LLM (GPT-3.5-0301) experiment is shown below:

> You are a helpful assistant that turns sentences into referring expressions such as those found in the RefCoco family of datasets. Turn the query into a referring expression. If the sentence does not explicitly have a target object infer one from the sentence's context Only return a single object. Examples of this include: find something to clean with on the left → broom on the left or Find me somewhere to put my cup of coffee → coffee table. If a referring expression is not required extract the object, e.g., Drive to the chair → "chair". Assume all queries refer to images that contain a semi-panoramic first-person view from an agent. Give ONLY the referring expression as the output. Query: INSERT_QUERY_HERE

The resulting referring expressions from this prompt can be seen in Tables 5, 6, 7, 8, 4. We note that in some instances the LLM failed to return a referring expression. Instead, we provide the result in the OFA +LLM expression column.

## 6.3 Full Results

All the prompts used in the classroom environment for testing the two different input representations A & B are presented in Table 4.

Table 4: Scene-05: Classroom

| Category | Sentence | A | B | OFA | OFA + LLM Expression | OFA+ LLM |
|----------|----------|---|---|-----|----------------------|----------|
| Generic | Go to the backpack | Pass | Pass | Fail | Backpack | Fail |
| Generic | Move towards the backpack | Pass | Pass | Fail | Backpack | Fail |
| Generic | Drive to the backpack | Pass | Pass | Fail | Backpack | Fail |
| Generic | Run towards the backpack | Pass | Pass | Fail | Backpack | Fail |
| Generic | Go to the cone | Pass | Pass | Fail | Cone | Fail |
| Generic | Go to the conical traffic delineator | Pass | Pass | Fail | Conical traffic delineator | Pass |
| Generic | Go to the trash can | Pass | Pass | Fail | Trash can | Pass |

| | | | | | | |
|---|---|---|---|---|---|---|
| Generic | Go to the whiteboard | Pass | Pass | Pass | Whiteboard | Pass |
| Generic | Proceed to the broom | Pass | Pass | Pass | Broom | Pass |
| Generic | Trek towards the wagon | Pass | Pass | Pass | Wagon | Pass |
| Generic | Find paper towels | Pass | Pass | Fail | Paper towels | Fail |
| Generic | Go to the outlet | Pass | Pass | Fail | Power outlet | Pass |
| Specific | Go to the red backpack | Pass | Pass | Fail | Red backpack | Fail |
| Specific | Go to the black backpack | Pass | Pass | Pass | Black backpack | Pass |
| Specific | Navigate to the backpack on the left | Pass | Pass | Fail | Backpack on the left | Fail |
| Specific | Drive to the backpack on the right | Pass | Pass | Fail | Backpack on the right | Fail |
| Specific | Go to the whiteboard in front of you | Pass | Pass | Pass | Whiteboard in front of you | Pass |
| Specific | Move to the whiteboard on your right | Pass | Pass | Fail | Whiteboard on your right | Fail |
| Specific | Move to the whiteboard on the right | Pass | Fail | Fail | Whiteboard on the right | Fail |
| Specific | Go to the backpack on your right | Pass | Pass | Fail | Backpack on your right | Pass |
| Specific | Walk to the backpack on the left | Pass | Fail | Fail | Backpack on the left | Pass |
| Specific | Go to the tv on the right | Pass | Fail | Pass | TV on the right | Pass |
| Specific | Go to the orange cone on your right | Fail | Pass | Fail | Orange cone on your right | Pass |
| Specific | Go to middle outlet | Pass | Fail | Fail | Middle outlet | Fail |
| Relational | Go to the backpack that is to the right of the red backpack | Pass | Pass | Fail | Backpack to the right of the red backpack | Fail |
| Relational | Drive to the backpack that is to the left of the black backpack | Pass | Fail | Fail | Backpack to the left of the black backpack | Fail |
| Relational | Walk to the bag that is next to the black bag | Fail | Fail | Fail | The bag next to the black bag | Fail |
| Relational | Move towards the backpack under the whiteboard | Pass | Pass | Fail | Backpack under whiteboard | Pass |
| Relational | Walk to the backpack on the chair | Pass | Pass | Fail | Backpack on the chair | Fail |
| Relational | Go to the chair with the backpack | Pass | Fail | Fail | Chair with the backpack | Fail |
| Relational | Walk to the backpack on top of the chair | Pass | Fail | Fail | Backpack on top of the chair | Fail |
| Relational | Run to the rightmost backpack | Pass | Pass | Fail | Rightmost backpack | Fail |
| Relational | Walk to the leftmost backpack | Pass | Pass | Fail | Leftmost backpack | Fail |
| Relational | Go to the middle chair | Pass | Fail | Fail | Middle chair | Fail |
| Relational | Go to the leftmost backpack on your right | Fail | Pass | Pass | Leftmost backpack on your right | Pass |
| Relational | Go to the middle chair in the row of chairs | Pass | Pass | Fail | Middle chair row of chairs | Fail |
| Relational | Go to the backpack on the left of the cone | Pass | Fail | Fail | Backpack on the left of the cone | Fail |
| Relational | Go to the cone to the left of the backpack | Pass | Pass | Fail | Cone to the left of the backpack | Fail |
| Relational | Go to the second chair from the left | Pass | Fail | Fail | Second chair from the left | Fail |
| Contextual | Go to somewhere I can sit down | Pass | Pass | Fail | Chair | Pass |
| Contextual | Find a place for me to rest | Pass | Fail | Fail | Couch | Fail |
| Contextual | Go to somewhere I can speak from | Pass | Fail | Fail | Podium | Pass |
| Contextual | Find a place to store cleaning supplies | Pass | Fail | Fail | Storage area | Fail |
| Contextual | Find me something to write on | Pass | Pass | Fail | Piece of paper | Pass |

| Contextual | My friend has a question. Go to somewhere you can explain the answer to him | Fail | Fail | Fail | My friend | Fail |
|---|---|---|---|---|---|---|
| Contextual | I spilled a lot of sand. Go find me something to pick up my mess | Pass | Pass | Fail | Something to pick up my mess → dustpan | Fail |
| Contextual | Walk to something where I can put my laptop in | Fail | Fail | Fail | Table to put laptop in | Fail |
| Contextual | I spilled water. Find me something to clean this up | Pass | Fail | Fail | Towel | Fail |
| Contextual | Go to somewhere I can google something | Pass | Pass | Fail | Computer on the left | Fail |
| Contextual | Go to somewhere I can charge my phone | Pass | Pass | Fail | Charging outlet | Pass |

Full results for the real-world experiments are shown in Tables 5, 6, 7, 8.

Table 5: Scene-01: Theater

| Category | Scene | Code | OD | WP | Path& Exec | OFA | OFA Expression | OFA+ LLM |
|---|---|---|---|---|---|---|---|---|
| Generic | Drive to the helmet | Pass | Pass | Pass | Pass | Fail | Helmet | Fail |
| Generic | Navigate to the mop | Pass | Pass | Pass | Pass | Fail | Mop | Fail |
| Generic | Go to the vacuum | Pass | Fail | Fail | Fail | Fail | Vaccum | Fail |
| Generic | Walk to the table | Pass | Pass | Pass | Pass | Fail | Table | Pass |
| Generic | Go to the fire extinguisher | Pass | Pass | Pass | Pass | Fail | Fire extinguisher | Fail |
| Generic | Walk to the saw | Pass | Fail | Fail | Fail | Fail | Saw | Fail |
| Specific | Go to the red backpack | Pass | Pass | Pass | Pass | Fail | Red backpack | Fail |
| Specific | Move to the orange bucket | Pass | Pass | Pass | Pass | Fail | Orange bucket | Fail |
| Specific | Drive to the chair on the right | Pass | Pass | Pass | Fail | Fail | Chair on the right | Fail |
| Specific | Walk to the white fan | Pass | Pass | Pass | Pass | Fail | White fan | Fail |
| Specific | Navigate to the black fan | Pass | Pass | Pass | Pass | Fail | Black fan | Fail |
| Specific | Navigate to the black speaker | Pass | Pass | Pass | Pass | Fail | Black speaker | Pass |
| Specific | Run to the red bag | Pass | Pass | Fail | Fail | Fail | Red bag | Fail |
| Specific | Move to the black chair infront of you | Pass | Pass | Pass | Pass | Fail | Black chair infront of you | Fail |
| Specific | Walk to the blue chair | Pass | Pass | Pass | Pass | Fail | Blue chair | Fail |
| Relational | Go to the chair with the black backpack on it | Pass | Pass | Pass | Pass | Pass | Chair with black backpack | Pass |
| Relational | Go to the chair with the helmet on it | Pass | Pass | Pass | Pass | Fail | Helmet Chair | Fail |
| Relational | Go to the backpack next to the chair | Pass | Fail | Fail | Fail | Fail | Backpack next to the chair | Fail |
| Relational | Go to the backpack infront of the ladder | Pass | Pass | Pass | Pass | Fail | Backpack infront of ladder | Fail |
| Relational | Run to the person on the ladder | Pass | Pass | Pass | Pass | Fail | Person on the ladder | Fail |
| Relational | Drive to the man sitting on the table | Pass | Pass | Pass | Pass | Fail | Man sitting on table | Fail |
| Relational | Go to the third chair from the left | Pass | Pass | Pass | Pass | Fail | Third chair from left | Fail |
| Contextual | Go to something that can carry water | Pass | Pass | Pass | Pass | Fail | Water container | Fail |
| Contextual | Navigate to something that can wash the floor | Fail | Fail | Fail | Fail | Fail | Mop | Fail |
| Contextual | My floor is dirty. Go to something that can fix this | Fail | Fail | Fail | Fail | Fail | Mop | Fail |
| Contextual | Go somewhere that I can throw something away in | Pass | Pass | Pass | Pass | Fail | Trash can | Pass |
| Contextual | Walk to something that put out a fire | Pass | Fail | Fail | Fail | Fail | Fire extinguisher | Fail |
| Contextual | Go to something that can cool me down | Pass | Fail | Fail | Fail | Fail | Fan | Fail |
| Contextual | Go to something white that will cool me down | Fail | Fail | Fail | Fail | Fail | Fan | Fail |
| Contextual | I am in the mood to listen to music. Go to something that can do that | Pass | Fail | Fail | Fail | Fail | Speaker with a music player | Fail |

Table 6: Scene-02: Lobby

| Category | Scene | Code | OD | WP | Path& Exec | OFA | OFA + LLM Expression | OFA+ LLM |
|---|---|---|---|---|---|---|---|---|
| Generic | Run to the trash bag | Pass | Pass | Pass | Pass | Pass | Trash bag | Pass |
| Generic | Navigate to the water fountain | Pass | Fail | Fail | Fail | Fail | Water fountain | Fail |
| Generic | Drive to the broom | Pass | Pass | Pass | Pass | Fail | Broom | Fail |
| Generic | Walk to the monitor | Pass | Pass | Fail | Fail | Fail | Monitor | Fail |
| Specific | Walk to the tiny monitor | Fail | Fail | Fail | Fail | Fail | Tiny monitor | Fail |
| Specific | Walk to the smallest monitor | Pass | Pass | Pass | Pass | Fail | Smallest monitor | Fail |
| Relational | Walk to the right most cone | Pass | Pass | Pass | Pass | Fail | Rightmost cone | Fail |
| Relational | Walk to the table with a can on it | Pass | Fail | Fail | Fail | Fail | Can on the table | Pass |
| Relational | Walk to the table next to the red backpack | Pass | Pass | Pass | Pass | Fail | Table next to red backpack | Pass |
| Relational | Go to the blue chair with the backpack on it | Pass | Pass | Pass | Pass | Pass | Blue chair with backpack | Pass |
| Relational | Walk to the leftmost table | Pass | Fail | Fail | Fail | Fail | Leftmost table | Fail |
| Relational | Go to the backpack closest to the shoes | Pass | Pass | Pass | Pass | Pass | Backpack closest to shoes | Pass |
| Relational | Walk to the shoe next to the red backpack | Pass | Pass | Pass | Pass | Fail | Shoes next to red backpack | Fail |
| Relational | Remove the trash bag from the floor | Fail | Fail | Fail | Fail | Fail | Trash bag on the floor | Pass |
| Relational | Walk to the table with the monitor on it | Fail | Fail | Fail | Fail | Pass | Table with monitor | Pass |
| Relational | Drive to the closest monitor to the table | Pass | Pass | Pass | Pass | Fail | Closest monitor to the table | Fail |
| Relational | Go to the table with the bottle on it | Fail | Fail | Fail | Fail | Fail | Table with the bottle | Fail |
| Relational | Go to the bottle on top of the table | Pass | Pass | Pass | Pass | Fail | Bottle on top of the table | Fail |
| Relational | Go to the tv closest to the person | Pass | Pass | Pass | Pass | Fail | TV closet to person | Fail |
| Relational | Go to the person with the blue shirt | Pass | Fail | Fail | Fail | Fail | Person with the blue shirt | Fail |
| Relational | Run to the chair with the blue coat on it | Pass | Fail | Fail | Fail | Fail | Run to the chair with the blue coat | Fail |
| Relational | Move to the backpack on the chair | Fail | Fail | Fail | Fail | Fail | Backpack on chair | Fail |
| Relational | Move to the black backpack on the chair | Fail | Fail | Fail | Fail | Fail | Black backpack on the chair | Fail |
| Relational | Drive to the chair with the backpack on it that is not red | Fail | Fail | Fail | Fail | Fail | Backpack on the chair | Fail |
| Contextual | Find something that can help a firefighter | Pass | Pass | Pass | Pass | Fail | Fire extinguisher | Fail |
| Contextual | Go to something that can clean a dirty floor | Fail | Fail | Fail | Fail | Fail | Mop | Fail |
| Contextual | Go get a drink of water | Fail | Fail | Fail | Fail | Fail | Water bottle | Fail |
| Contextual | Go to a place where I can watch a movie | Pass | Pass | Pass | Pass | Fail | Movie theater | Fail |
| Contextual | Drive to a place where I can watch a video | Fail | Fail | Fail | Fail | Fail | TV or screen | Fail |

Table 7: Scene-03: Outdoor

| Category | Scene | Code | OD | WP | Path& Exec | OFA | OFA + LLM Expression | OFA+ LLM |
|---|---|---|---|---|---|---|---|---|
| Generic | Wander to the fire hydrant | Pass | Pass | Pass | Pass | Fail | Fire hydrant | Fail |
| Generic | Step towards the gril | Pass | Fail | Fail | Fail | Pass | Grill | Pass |
| Generic | Walk to the skateboard | Pass | Pass | Pass | Pass | Fail | Skateboard | Fail |
| Generic | Walk to the bike | Pass | Pass | Fail | Fail | Fail | Bike | Fail |
| Generic | Walk to the bike | Pass | Pass | Pass | Pass | Fail | Bike | Fail |
| Generic | Go to the bike rack | Pass | Pass | Pass | Pass | Fail | Bike rack | Fail |
| Generic | Go to the sign | Pass | Pass | Fail | Fail | Fail | The sign | Fail |
| Generic | Go to the bench | Pass | Pass | Pass | Pass | Fail | Bench | Fail |
| Specific | Sashay to the stop sign | Pass | Pass | Pass | Pass | Fail | Stop sign | Fail |
| Specific | Go to the basketball hoop | Pass | Pass | Pass | Pass | Fail | Basketball hoop | Pass |
| Specific | Roam towards the blue car | Pass | Pass | Pass | Pass | Fail | Blue car | Pass |
| Specific | Trot towards the red bag | Pass | Pass | Pass | Pass | Fail | Red bag | Fail |
| Specific | Go to the red object | Fail | Fail | Fail | Fail | Fail | Red object | Fail |
| Relational | Proceed to the middle cone | Pass | Pass | Pass | Pass | Pass | Middle cone | Pass |
| Relational | Journey to the tree next to the backpack | Fail | Fail | Fail | Fail | Fail | Tree next to the backpack | Pass |
| Relational | Trek to the backpack by the tree | Pass | Pass | Pass | Pass | Fail | Backpack by the tree | Fail |
| Contextual | A firefighter needs water. Walk to a source of water | Pass | Fail | Fail | Fail | Fail | Source of water | Fail |
| Contextual | Head towards something that can help firefighters | Pass | Pass | Pass | Pass | Fail | Fire hydrant | Pass |
| Contextual | You are a dog that needs to mark its territory. Go find a place to do this | Pass | Pass | Pass | Pass | Fail | Fire hydrant | Fail |
| Contextual | You are carrying trash, Find somewhere to dump it | Pass | Pass | Pass | Pass | Fail | Dumpster | Pass |
| Contextual | Find me something to do a kickflip on | Pass | Pass | Pass | Pass | Fail | Skateboard | Fail |
| Contextual | I want to shoot some hoops. Take me there | Fail | Fail | Fail | Fail | Fail | Basketball court | Fail |
| Contextual | Move towards a faster mode of transportation | Pass | Pass | Pass | Pass | Fail | Sorry I can't generate a referring expression | Fail |
| Contextual | Head to the fastest mode of transportation | Pass | Pass | Pass | Pass | Pass | Fastest mode of transportation | Pass |

Table 8: Scene-04: Courtyard

| Category | Scene | Code | OD | WP | Path& Exec | OFA | OFA + LLM Expression | OFA+ LLM |
|---|---|---|---|---|---|---|---|---|
| Generic | Run to the door | Pass | Pass | Pass | Pass | Fail | Door | Fail |
| Generic | Run towards the backpack | Pass | Pass | Pass | Pass | Fail | Backpack | Fail |
| Generic | Drive to the wagon | Pass | Pass | Pass | Pass | Fail | Wagon | Pass |
| Generic | Navigate to the stairs | Pass | Pass | Pass | Pass | Pass | Stairs | Pass |
| Specific | Proceed towards the garbage can on the right | Pass | Pass | Pass | Pass | Fail | Garbage can on the right | Fail |
| Specific | Stroll to the recycle bin on the left | Pass | Pass | Fail | Fail | Fail | Recycle bin on the left | Fail |
| Specific | Sprint to the picnic table | Pass | Pass | Pass | Pass | Fail | Picnic table | Fail |
| Relational | Go to the bench with water container on it | Pass | Pass | Pass | Pass | Pass | Bench with water container | Pass |
| Relational | Walk to the bench with nothing on it | Pass | Pass | Pass | Pass | Fail | Bench with nothing on it | Pass |
| Relational | Proceed to the bench with most objects | Fail | Fail | Fail | Fail | Fail | Bench with the most objects | Pass |
| Relational | Move towards the backpack farthest from a bench | Pass | Fail | Fail | Fail | Fail | Backpack farthest away from bench | Fail |
| Relational | Head towards the middle cone | Fail | Fail | Fail | Fail | Fail | Middle cone in row of cones | Fail |
| Relational | Head towards the middle cone in the row of cones | Pass | Pass | Pass | Pass | Fail | Middle cone in row of cones | Fail |
| Relational | Go to the table with only one chair | Fail | Fail | Fail | Fail | Pass | Table with only one chair | Pass |
| Relational | Go to the table with only one chair. There are multiple groups of chairs around multiple tables | Pass | Pass | Pass | Pass | Pass | Table with only one chair | Pass |
| Relational | Step towards the column closest to the cart | Pass | Pass | Pass | Pass | Fail | Column closest to the cart | Pass |
| Relational | Move to the largest group of benches | Pass | Pass | Pass | Pass | Fail | Largest group of benches | Fail |
| Relational | Walk towards the table with the umbrella | Pass | Pass | Pass | Pass | Fail | Umbrella on the table | Pass |
| Relational | Walk towards the table with the yellow umbrella | Pass | Pass | Pass | Pass | Fail | Table with yellow umbrella | Fail |
| Relational | Drive to a table without any chairs | Pass | Pass | Pass | Pass | Fail | Table without chairs | Fail |
| Relational | Walk to black table with six chairs | Fail | Fail | Fail | Fail | Fail | Black table with six chairs | Fail |
| Relational | Walk to the table with the most chairs | Pass | Pass | Pass | Pass | Fail | Table with the most chairs | Fail |
| Relational | Navigate to the table with the backpack | Fail | Fail | Fail | Fail | Fail | Backpack on the table | Fail |
| Contextual | Go to the nearest entrance to the building | Fail | Fail | Fail | Fail | Fail | Nearest entrance to the building | Fail |
| Contextual | Go to something that you would hold open for someone elderly | Pass | Fail | Fail | Fail | Fail | Door | Fail |
| Contextual | Go to something that will make it easier to carry heavy luggage | Pass | Fail | Fail | Fail | Pass | Cart | Pass |
| Contextual | Go to somewhere I can eat my lunch | Pass | Fail | Fail | Fail | Fail | Lunch table | Pass |
| Contextual | Go up to the second floor | Fail | Fail | Fail | Fail | Fail | Second floor | Fail |
| Contextual | Go find something to climb | Fail | Fail | Fail | Fail | Fail | Rock wall | Fail |
| Contextual | Run upstairs | Pass | Pass | Pass | Pass | Fail | Stairs | Fail |
| Contextual | Find me somewhere to park my bike | Pass | Pass | Pass | Pass | Fail | Bike rack | Pass |

## 6.4 Code Prompt

Navigation Prompt for the concatenated single image input. We leverage the original prompts presented in [14].

```python
import math

class ImagePatch:
    """A Python class containing a crop of an image centered around a
        particular object, as well as relevant information.
    Attributes
    ----------
    cropped_image : array_like
        An array-like of the cropped image taken from the original image.
    left, lower, right, upper : int
        An int describing the position of the (left/lower/right/upper)
            border of the crop's bounding box in the original image.
    frame: name of camera frame

    Methods
    -------
    find(object_name: str)->List[ImagePatch]
        Returns a list of new ImagePatch objects containing crops of the
            image centered around any objects found in the
        image matching the object_name.
    exists(object_name: str)->bool
        Returns True if the object specified by object_name is found in the
            image, and False otherwise.
    verify_property(property: str)->bool
        Returns True if the property is met, and False otherwise.
    best_text_match(option_list: List[str], prefix: str)->str
        Returns the string that best matches the image.
    simple_query(question: str=None)->str
        Returns the answer to a basic question asked about the image. If no
            question is provided, returns the answer to "What is this?".
    llm_query(question: str, long_answer: bool)->str
        References a large language model (e.g., GPT) to produce a response
            to the given question. Default is short-form answers, can be
            made long-form responses with the long_answer flag.
    compute_depth()->float
        Returns the median depth of the image crop.
    crop(left: int, lower: int, right: int, upper: int)->ImagePatch
        Returns a new ImagePatch object containing a crop of the image at
            the given coordinates.
    """

    def __init__(self, image, left: int = None, lower: int = None, right:
        int = None, upper: int = None, frame = None):
        """Initializes an ImagePatch object by cropping the image at the
            given coordinates and stores the coordinates as
        attributes. If no coordinates are provided, the image is left
            unmodified, and the coordinates are set to the
        dimensions of the image.
        Parameters
        -------
        image : array_like
            An array-like of the original image.
        left, lower, right, upper : int
```

```
                An int describing the position of the (left/lower/right/upper)
                    border of the crop's bounding box in the original image.
        """
        if left is None and right is None and upper is None and lower is
            None:
            self.cropped_image = image
            self.left = 0
            self.lower = 0
            self.right = image.shape[2] # width
            self.upper = image.shape[1] # height
        else:
            self.cropped_image = image[:, lower:upper, left:right]
            self.left = left
            self.upper = upper
            self.right = right
            self.lower = lower

        self.width = self.cropped_image.shape[2]
        self.height = self.cropped_image.shape[1]

        self.horizontal_center = (self.left + self.right) / 2
        self.vertical_center = (self.lower + self.upper) / 2

        self.frame = frame

    def find(self, object_name: str) -> List[ImagePatch]:
        """Returns a list of ImagePatch objects matching object_name
            contained in the crop if any are found.
        Otherwise, returns an empty list.
        Parameters
        ----------
        object_name : str
            the name of the object to be found

        Returns
        -------
        List[ImagePatch]
            a list of ImagePatch objects matching object_name contained in
                the crop

        Examples
        --------
        >>> # return the foo
        >>> def execute_command(image) -> List[ImagePatch]:
        >>> image_patch = ImagePatch(image)
        >>> foo_patches = image_patch.find("foo")
        >>> return foo_patches
        """
        return find_in_image(self.cropped_image, object_name)

    def exists(self, object_name: str) -> bool:
        """Returns True if the object specified by object_name is found in
            the image, and False otherwise.
        Parameters
        -------
        object_name : str
            A string describing the name of the object to be found in the
                image.
```

```python
    Examples
    -------
    >>> # Are there both foos and garply bars in the photo?
    >>> def execute_command(image)->str:
    >>> image_patch = ImagePatch(image)
    >>> is_foo = image_patch.exists("foo")
    >>> is_garply_bar = image_patch.exists("garply bar")
    >>> return bool_to_yesno(is_foo and is_garply_bar)
    """
    return len(self.find(object_name)) > 0

def verify_property(self, object_name: str, visual_property: str) ->
    bool:
    """Returns True if the object possesses the visual property, and
        False otherwise.
    Differs from 'exists' in that it presupposes the existence of the
        object specified by object_name, instead checking whether the
        object possesses the property.
    Parameters
    -------
    object_name : str
        A string describing the name of the object to be found in the
            image.
    visual_property : str
        A string describing the simple visual property (e.g., color,
            shape, material) to be checked.

    Examples
    -------
    >>> # Do the letters have blue color?
    >>> def execute_command(image) -> str:
    >>> image_patch = ImagePatch(image)
    >>> letters_patches = image_patch.find("letters")
    >>> # Question assumes only one letter patch
    >>> return bool_to_yesno(letters_patches[0].verify_property("letters
        ", "blue"))
    """
    return verify_property(self.cropped_image, object_name, property)

def best_text_match(self, option_list: List[str]) -> str:
    """Returns the string that best matches the image.
    Parameters
    -------
    option_list : str
        A list with the names of the different options
    prefix : str
        A string with the prefixes to append to the options

    Examples
    -------
    >>> # Is the foo gold or white?
    >>> def execute_command(image)->str:
    >>> image_patch = ImagePatch(image)
    >>> foo_patches = image_patch.find("foo")
    >>> # Question assumes one foo patch
    >>> return foo_patches[0].best_text_match(["gold", "white"])
    """
    return best_text_match(self.cropped_image, option_list)
```

```python
    def simple_query(self, question: str = None) -> str:
        """Returns the answer to a basic question asked about the image. If
            no question is provided, returns the answer
        to "What is this?". The questions are about basic perception, and
            are not meant to be used for complex reasoning
        or external knowledge.
        Parameters
        -------
        question : str
            A string describing the question to be asked.

        Examples
        -------

        >>> # Which kind of baz is not fredding?
        >>> def execute_command(image) -> str:
        >>> image_patch = ImagePatch(image)
        >>> baz_patches = image_patch.find("baz")
        >>> for baz_patch in baz_patches:
        >>> if not baz_patch.verify_property("baz", "fredding"):
        >>> return baz_patch.simple_query("What is this baz?")

        >>> # What color is the foo?
        >>> def execute_command(image) -> str:
        >>> image_patch = ImagePatch(image)
        >>> foo_patches = image_patch.find("foo")
        >>> foo_patch = foo_patches[0]
        >>> return foo_patch.simple_query("What is the color?")

        >>> # Is the second bar from the left quuxy?
        >>> def execute_command(image) -> str:
        >>> image_patch = ImagePatch(image)
        >>> bar_patches = image_patch.find("bar")
        >>> bar_patches.sort(key=lambda x: x.horizontal_center)
        >>> bar_patch = bar_patches[1]
        >>> return bar_patch.simple_query("Is the bar quuxy?")
        """
        return simple_query(self.cropped_image, question)

    def compute_depth(self):
        """Returns the median depth of the image crop
        Parameters
        ----------
        Returns
        -------
        float
            the median depth of the image crop

        Examples
        --------
        >>> # the bar furthest away
        >>> def execute_command(image)->ImagePatch:
        >>> image_patch = ImagePatch(image)
        >>> bar_patches = image_patch.find("bar")
        >>> bar_patches.sort(key=lambda bar: bar.compute_depth())
        >>> return bar_patches[-1]
        """
        depth_map = compute_depth(self.cropped_image)
        return depth_map.median()
```

```python
    def crop(self, left: int, lower: int, right: int, upper: int) ->
        ImagePatch:
        """Returns a new ImagePatch cropped from the current ImagePatch.
        Parameters
        -------
        left, lower, right, upper : int
            The (left/lower/right/upper)most pixel of the cropped image.
        -------
        """
        return ImagePatch(self.cropped_image, left, lower, right, upper,
            self.frame)

    def overlaps_with(self, left, lower, right, upper):
        """Returns True if a crop with the given coordinates overlaps with
            this one,
        else False.
        Parameters
        ----------
        left, lower, right, upper : int
            the (left/lower/right/upper) border of the crop to be checked

        Returns
        -------
        bool
            True if a crop with the given coordinates overlaps with this one
                , else False

        Examples
        --------
        >>> # black foo on top of the qux
        >>> def execute_command(image) -> ImagePatch:
        >>> image_patch = ImagePatch(image)
        >>> qux_patches = image_patch.find("qux")
        >>> qux_patch = qux_patches[0]
        >>> foo_patches = image_patch.find("black foo")
        >>> for foo in foo_patches:
        >>> if foo.vertical_center > qux_patch.vertical_center
        >>> return foo
        """
        return self.left <= right and self.right >= left and self.lower <=
            upper and self.upper >= lower

def best_image_match(list_patches: List[ImagePatch], content: List[str],
    return_index=False) -> Union[ImagePatch, int]:
    """Returns the patch most likely to contain the content.
    Parameters
    ----------
    list_patches : List[ImagePatch]
    content : List[str]
        the object of interest
    return_index : bool
        if True, returns the index of the patch most likely to contain the
            object

    Returns
    -------
    int
        Patch most likely to contain the object
```

```
        """
        return best_image_match(list_patches, content, return_index)

def distance(patch_a: ImagePatch, patch_b: ImagePatch) -> float:
    """
    Returns the distance between the edges of two ImagePatches. If the
        patches overlap, it returns a negative distance
    corresponding to the negative intersection over union.

    Parameters
    ----------
    patch_a : ImagePatch
    patch_b : ImagePatch

    Examples
    --------
    # Return the qux that is closest to the foo
    >>> def execute_command(image):
    >>> image_patch = ImagePatch(image)
    >>> qux_patches = image_patch.find('qux')
    >>> foo_patches = image_patch.find('foo')
    >>> foo_patch = foo_patches[0]
    >>> qux_patches.sort(key=lambda x: distance(x, foo_patch))
    >>> return qux_patches[0]
    """
    return distance(patch_a, patch_b)

def bool_to_yesno(bool_answer: bool) -> str:
    return "yes" if bool_answer else "no"

def coerce_to_numeric(string):
    """
    This function takes a string as input and returns a float after removing
        any non-numeric characters.
    If the input string contains a range (e.g. "10-15"), it returns the
        first value in the range.
    """
    return coerce_to_numeric(string)

### Nav Client
"""
Navigates an agent to an object in an imgae given its center coordinates
Parameters
    ----------
    x : x coordinate of the center of the object
    y : y coordinate of the center of the object
"""
 Examples
        -------
        >>> # Go to the blue foo.
        >>> def execute_command(image)
        >>> image_patch = ImagePatch(image)
        >>> foo_patches = image_patch.find("foo")
        >>> # Verify visual property
        >>> blue_color_patches = []
        >>> for foo_patch in foo_patches:
```

```
        >>> if verify_property(blue, "color")
        >>> blue_color_patches.append(foo_patch)
        >>> inputs = (blue_color_patches[0].horizontal_center,
            blue_color_patches[0].vertical_center)
        >>> return {'function': 'nav_function', 'inputs': inputs, 'box': [
            blue_color_patches[0].left, blue_color_patches[0].lower,
            blue_color_patches[0].right, blue_color_patches.upper]}
        """
def navigate_to_object(double x, double y):

"""

Write a function using Python and/or the ImagePatch class (above) that
    could be executed to provide an answer to the query by calling one of
    the functions in the Nav Client (navigate_to_object).
Note using the ImagePatch is not required for all queries. If the query is
    not relevant to navigation, return None for the function and a string
    describing the problem.

Consider the following guidelines:
- Use base Python (comparison, sorting) for basic logical operations, left/
    right/up/down, math, etc.
- Use the llm_query function to access external information and answer
    informational questions not concerning the image.
- All properties such as color should be verified using the verify_property
     function. "go to the blue foo" implies "go to foo if foo is blue"
- The output of this function should be a dict {function: 'function in the
    nav client', inputs: 'inputs to the chosen nav client function', box: [
    left, lower, right, upper]}, or for an error {function: 'None', error:
    'message describing the problem'}
- If more than one object is found pick the best match

Query: INSERT_QUERY_HERE
```

