# OpenReview forum: "Tell Me Where to Go: A Composable Framework for Context-Aware Embodied Robot Navigation"
_robot-learning.org/CoRL/2023/Conference — CoRL 2023 Poster_

### Official Review · Reviewer_DATK · 2023-07-17

**Confidence:** 5
**Originality:** Fair
**Technical Quality:** Good
**Clarity Of Presentation:** Good
**Impact:** 2

**Recommendation:**

Weak Reject: I recommend rejecting the paper, but will not argue for my recommendation if the majority of other reviewers have a different opinion.

**Review:**

Strengths
---
- Grounding LLMs in robotic experience is indeed fundamental to their applications in robotics, and this paper proposes a good recipe of grounding using object detections and simple localization. While it is not clear if the proposed object-centric representation would scale to more complex settings, it is very flexible and easy to integrate with off-the-shelf LLMs.
- The analysis of visual representations for grounding VLMs is quite interesting and may be useful for people working in related areas.
- The discussion on related works seems very thorough and complete.

***
Weaknesses
---
- The presentation of the method seems incomplete and the system overview is very hard to understand. It appears that the authors use an object centric representation, but
	- It is not clear how these objects are grounded in the visual inputs of the robot. Presumably, this would require an open-vocabulary detector (e.g., GLIP, DETIC etc.) but this is not discussed or described anywhere in the paper. In its current form, the paper is not reproducible.
	- How are the “skills”, or the “navigate_to_object” skills trained and executed? It seems that the skills are conditioned on 2D positions of the goals, and in the presence of a 3D map (as shown in Figure 3), this should be fairly straightforward using simple motion planning, but this process is not described at any point. The authors should include a clear set of assumptions and problem setup for the reader to contextualize the findings.

- As an empirical paper, the experiments lack rigor, both in terms of the hypotheses tested and lack of comparisons to any standard baselines.
	- Section 4 studies 2 questions: (i) what the right visual input is (stitched panorama v/s frame stacking) and (ii) breakdown of success/failure in terms of perception, planning etc. and the authors report “100% success rate for code generation” and “100% success rate in following all planned paths”. While this is somewhat informative, it is very hard to understand the performance of the system without any competent baselines also included. There is a variety of prior work on this topic (many of which are cited and discussed in Section 2: code-as-policies style system, LM-Nav etc.) that would form useful baselines; the authors could also include simpler baselines that do not use LLMs, methods using grounded VQA models (BLIP2 etc.) and ablations of various system components to truly understand the design space. In its current form, it is not clear whether the method is competent or the task too easy.
	- It is also concerning that the only experiment discussed in detail/visualized (Figure 5, 6) do not actually have any control component and are simple VQA-style queries. Given the task is not benchmarked against baselines, this further raises the concerns regarding the validity of numbers reported. For the task shown in Figure 5 and 6, assuming there are low level “go to X”  skills, it is not clear why an off-the-shelf VQA model would not solve the task either.

- Reading the discussion in L221-236, it appears that the authors’ stated limitations have less to do with the proposed method, and more likely an effect of incomplete “prompts” to the LLM. It appears to me that the pattern “find image patch, sort and count it” appears too often in the prompt and the outputs are heavily overfit to that kind of reasoning (see L225-232, Figure 5). (This is not strictly a weakness of the method, and in fact suggests that these limitations can be very easily patched)

- An implicit assumption (which should be stated more prominently) in the paper seems to be that navigation skills of the type “go to X” are sufficient for solving this task. This is a very strong assumption and severely limits the applications of the method; since the interface to this skill is just a 2D position, it also limits the reasoning to reaching line-of-sight goals, and cannot do more interesting things such as “reach behind the couch” or “drive slowly to the left of the pedestrian” etc.
	- In L188-189, the authors say “We find that we are able to successfully generate code for navigational plans using a variety of verbs e.g., walk, go, drive, run, etc” but this seems FALSE. How are verbs handled by the low-level policy if all it can do is “go to position X”? A clarification on these aspects would be very useful.


**Quality Of The Limitations Section:**

Additional details required

**Questions For Rebuttal:**

Please see weaknesses section above on detailed concerns.
- Lack of experimental rigor and baselines
- Better discussion of assumptions and limitations
- Missing key implementation details and reproducibility concerns

**Robotics Focus:**

Highly relevant to robotics but no hardware experiments

**Summary Of Paper:**

The authors propose a system for using LLMs for context-aware reasoning for real-world robot navigation, using code generation. The visual observations are grounded using the abstraction of objects and 2D positions, and the LLM reasons over this abstraction and a user-provided instruction. The authors show qualitative and quantitative analysis, including real-world deployment.


**Summary Of Recommendation:**

The paper presents a simple yet effective idea in using natural language context to guide robotic navigation by grounding visual observations in text. However, I cannot recommend acceptance in its current form due to missing key implementation details and incomplete evaluations. I encourage the authors to revise relevant sections, and I am open to reconsidering my recommendation after the author rebuttal/discussion phase.

---

Updated my recommendation to WR after rebuttal phase discussion.

---

### Official Review · Reviewer_9PgY · 2023-07-20

**Confidence:** 4
**Originality:** Fair
**Technical Quality:** Good
**Clarity Of Presentation:** Good
**Impact:** 3

**Recommendation:**

Weak Accept: I recommend accepting the paper, but will not argue for my recommendation if the majority of other reviewers have a different opinion.

**Review:**

This paper presents an expressive framework for executing natural language commands in navigation domains. Code is a promising intermediate representation for robotic tasks as it allows engineers to incorporate domain knowledge and state of the art modules. Although the idea has shown to be useful in computer vision domains (ViperGPT, [14]), it is a useful contribution to apply the method in robotic domains where actions also need to be executed on the real robot.

A key strength of this paper is the number of real world navigation experiments across a diverse set of domains. The results show that the generated Python code can successfully be executed to complete the task. The authors also do a good job analyzing which parts of the system lead to failures and identifying areas for improvement.

One weakness of this work is that there are no non-LLM baselines as comparisons. As such, for each of the command categories, it is difficult to interpret how good the reported metrics are. As stated, LLMs have the capacity to perform sophisticated reasoning tasks, but it is unclear how much this sophistication is needed for the tasks in this paper. They likely offer large performance gains for “contextual commands” (65.22% code generation success) and less for “generic commands” (100% code generation success). I do not think it’s necessary to have extensive baseline evaluations. However, it would strengthen the paper if we could tell how much the proposed framework outperforms a model that doesn’t use prior knowledge from LLMs, or a less expressive intermediate layer than Python code.

I also have a few concerns/questions regarding clarity which I believe can be addressed in the rebuttal period:
- In Tables 2 and 3, are the success rates over all examples or only the ones where the previous stage was successful? For example, does OD (%) only include trials where the code generation was successful?
- On line 214, there is a claim that the proposed framework can find better solutions than humans would. However, the only evidence to support this is a single example. This is indeed an interesting observation but needs further statistical analysis to support the claim. Without that, I would recommend a weaker claim that this observation should be investigated in future work.
- How were the sentences generated for the experiments? (Appendix Tables 5-8)
- Why do the contextual results in Table 1 appear better than the results in Table 2? Can this mainly be attributed to the small sample size?

----- Post-Rebuttal Update -----

I would like to thank the authors for taking the time to respond to my questions and concerns. The additional details are helpful in understanding the methods and results.

I also appreciate the addition of the OFA baseline but have additional concerns about the included results. In particular, the baseline does not appear to be a competitive baseline and has poor performance even for generic sentences which do not involve any complicated reasoning. In a response to other reviewers, the authors mention this is largely due to predicting the wrong bounding box. As such, it is a useful ablation of the Code + OD  components. But it is difficult to tease out if the baseline is failing primarily due to vision or from lacking prior knowledge included in the LLM.

I have updated my score to a weak accept. This work is promising and code generation is a potentially powerful approach to enable execution of complex language commands.

**Quality Of The Limitations Section:**

Limitations are addressed clearly

**Questions For Rebuttal:**

- Inclusion of a baseline as a way to calibrate performance results.
- Additional details about experimental setup and results (see above review).

**Robotics Focus:**

Sufficient demonstration on hardware

**Summary Of Paper:**

This paper presents a navigation framework that leverages background and common-sense knowledge. Human operators frequently express navigational goals in ways that require prior knowledge about the world. For example, “Get me something to hold the poster up”, would require the robot to recognize that “tape” or “tacks” would accomplish the task. Large Language Models (LLMs) have been successful at this type of reasoning, but they lack the grounding needed to interact with the physical world. The proposed framework addresses this challenge by using advances in code generation to define an intermediate layer between the LLM and grounding/action primitives. Specifically, a Python API is proposed that can perform operations such as object recognition and navigation commands. The LLM generates code using this API such that the resulting program, when executed, will complete the task. The framework is evaluated in four diverse domains using a variety of natural language expressions.

**Summary Of Recommendation:**

I weakly recommend rejecting this paper in the current state. Although the authors show success on a real system, it is difficult to tell what successes come from the proposed framework. However, I believe this issue can be addressed and look forward to discussion during the rebuttal period.

---

### Official Review · Reviewer_Gcpd · 2023-07-20

**Confidence:** 3
**Originality:** Good
**Technical Quality:** Good
**Clarity Of Presentation:** Good
**Impact:** 3

**Recommendation:**

Weak Accept: I recommend accepting the paper, but will not argue for my recommendation if the majority of other reviewers have a different opinion.

**Review:**

The proposed framework is a novel approach that bridges the gap between large language models (LLMs) and robot navigation by providing an intermediate layer in the form of Python code. This code allows robots to understand natural language descriptions of the environment and associate them with physical objects. The results of the experiments conducted in four different environments and command classes on a mobile robot showed that the framework had an overall success rate of 90% accuracy across 50 different sentences in controlled experiments and 70% execution accuracy in real-world environments, which is significantly higher than the baselines. The proposed framework was evaluated across four different environments and command classes on a mobile robot, and the results showed that the framework had an overall success rate of 78.07% for code generation, 63.16% for object detection, 58.77% for waypoint navigation, and 57.89% for path and execution. The proposed method outperformed the baselines in terms of accuracy and execution, achieving 90% accuracy across 50 different sentences in controlled experiments and 70% execution accuracy in real-world environments. However, there were some failures due to the wrong object being detected or the object not being detected at all, as well as the vision system detecting further out than the map and projections missing the correct voxel on the map.

However, the paper also has some limitations and weaknesses. The object detector has some failures due to incorrect object detection or objects not being detected at all. This is evidenced by the paper, which states that the object detector (GLIP) was successful in detecting objects, however, there were some failures due to the wrong object being detected or the object not being detected at all. Additionally, the paper also mentions that the waypoint projection step had some failures due to the vision system detecting further out than the map and projections missing the correct voxel on the map. These failures made the approach hard to deploy in the real world. Besides, the paper lacks many descriptions of how the authors constructs the framework and evaluated the results. For example, there are many "success rates" invarious tables, but the authors didn't give how they were defined, weakening the arguments of the paper.

**Quality Of The Limitations Section:**

Limitations are addressed clearly

**Questions For Rebuttal:**

1. The success for experiment is not clearly stated, the authors should be explicit about how they evaluate their approaches, and if it is based on human questionaire, clearly state how there is a clear rule to obtain reproducible results.

2. What are the success metrics for sub modules like Code Generation, OD, WP, P&E? They shall be clearly stated in the paper.

3. How did the authors get the map (m), how does the authors determine if the robot is able to navigate inside the terrain (like area with stairs), what is the data format of the map and how it is fed into the model?

4. The paper also lacks key comparison with classic method such as a hierarchical object inference + object navigation policy.

**Robotics Focus:**

Sufficient demonstration on hardware

**Summary Of Paper:**

This paper presents a composable framework for robot navigation that leverages language models (LLMs), state-of-the-art object detectors, and classical robotic planning algorithms to perform zero-shot natural language based navigation in four unique environments. The framework addresses both the grounding and transparency issues of current LLMs, and requires a minimal uplink for the robot since all of the planning, mapping, and localization is performed onboard. The authors evaluate different 2D input representations to determine an effective way to extract spatial and conceptual knowledge from LLMs, and perform extensive real-world experiments in a variety of environments to show that the framework is able to navigate to landmarks based on natural language. The results of the experiments showed that the system was able to successfully navigate in a variety of environments and that the input representation was able to accurately capture the necessary information for navigation

**Summary Of Recommendation:**

This paper is a good embodied extension of ViperGPT: Visual Inference via Python Execution for Reasoning, while the contribution is limited to the adaptation and analysis of how the approach can work in the navigation setting. The demos are promising, but more justifications of the approach capability and how the success is due to the authors contribution rather than the pure in-context learning is to be demonstrated.

---

### Official Review · Reviewer_cxP7 · 2023-07-20

**Confidence:** 4
**Originality:** Very Good
**Technical Quality:** Good
**Clarity Of Presentation:** Good
**Impact:** 4

**Recommendation:**

Weak Accept: I recommend accepting the paper, but will not argue for my recommendation if the majority of other reviewers have a different opinion.

**Review:**

Pros:
1. Innovative approach, connects language models to embodied navigation in unseen environments
2. Robot writes navigation code, this is also innovative
3. Good discussion on limitations. This seems like primarily a multimodal embodied reasoning but authors have done the best to map it to language.

Cons:
1. One complaint I have with this paper is that many of the components are referenced but not explained, such as the algorithm for avoiding terrains. I'm curious but don't have the time to go look at the paper, and could've used a one sentence summary. Similarly for details of intermediate layer.
2. This is not quite a robot learning paper in that nothing is trained, but is a prompt engineering paper. That's ok, I guess.
Ablations on design choices is not included.
3. Not many ablations for design choices



**Quality Of The Limitations Section:**

Limitations are addressed clearly

**Questions For Rebuttal:**

1. Could you add some reasoning as to why you designed the system this way and not any other way. What were alternatives considered? eg why concatenation of camera views as against language description?


**Robotics Focus:**

Sufficient demonstration on hardware

**Summary Of Paper:**

It's a code as policies type paper for navigation. Given an instruction, robot writes a plan on where to go, and then this is executed. Inputs are front, left and right images. Four types of instructions are tested: generic, specific, relational and contextual.

**Summary Of Recommendation:**

This is a paper that tries an innovative approach at robot navigation using LLMs. Unfortunately it is not a learning paper but a prompting paper. But theyve designed very good engineering around it, that more or less massages the fact that LLMs can't see. Unfortunately, many details of that engineering is taken from other works but not explained clearly in the paper, so it seems a little bit like magic to a reader who hasnt read the references. Small explanations of what's going on would be useful. Results are interesting, failure modes and limitations are thoroughly explained.

---

### Decision · Program_Chairs · 2023-08-30

**Decision:**

Accept (Poster)

**Comment:**

The paper proposes a system that uses LLMs for context-aware commonsense reasoning for real-world robot navigation, using code generation. The visual observations are grounded using the abstraction of objects and 2D positions, and the LLM reasons over this abstraction and a user-provided instruction, to generate Python code using a given API such that the resulting program, when executed, will complete the task. The authors show qualitative and quantitative analysis in four domains, including real-world deployment.

Reviewers agree that the proposed approach is flexible (DATK), that code presents a promising intermediate representation for navigation tasks (9PgY, cxP7), and that a key strength of the presented system is that it has been tested across several real-world domains (9PgY). However, reviewers also raised important concerns, a primary one being that there are missing comparisons to competitive baselines (DATK, 9PgY, Gcpd) missing component ablations (cxP7), and missing details for clarity and reproducibility.

Post-rebuttal, the manuscript has improved substantially in clarity, and now includes a baseline comparison to OFA (Wang et al.). Reviewer ratings have been updated to: weak accept, weak accept, weak accept, weak reject. Nevertheless, reviewers point out that the new comparison is not necessarily a fair evaluation (DATK, 9PgY) since the proposed pipeline uses an LLM + GLIP, while the baseline is only end-to-end OFA (no LLM). The relatively large gap in performance between the proposed system (71%) and baseline (12%) corroborates this. I would strongly recommend strengthening the baseline to improve the evaluations for the revision.